# Development of an Antigen Capture Lateral Flow Immunoassay for the Detection of *Burkholderia pseudomallei*

**DOI:** 10.3390/diagnostics14101033

**Published:** 2024-05-16

**Authors:** Teerapat Nualnoi, Paweena Wongwitwichot, Siriluk Kaewmanee, Pornchanan Chanchay, Nattapong Wongpanti, Tossapol Ueangsuwan, Rattikarn Siangsanor, Wannittaya Chotirouangnapa, Tanatchaporn Saechin, Suwanna Thungtin, Jidapa Szekely, Chaiyawan Wattanachant, Vannarat Saechan

**Affiliations:** 1Department of Pharmaceutical Technology, Faculty of Pharmaceutical Sciences, Prince of Songkla University, Hat Yai 90110, Songkhla, Thailandbanktossapol.u@gmail.com (T.U.);; 2Drug Delivery System Excellence Center (DDSEC), Faculty of Pharmaceutical Sciences, Prince of Songkla University, Hat Yai 90110, Songkhla, Thailand; paweena.w@psu.ac.th; 3Department of Pharmaceutical Chemistry, Faculty of Pharmaceutical Sciences, Prince of Songkla University, Hat Yai 90110, Songkhla, Thailand; 4Faculty of Veterinary Science, Prince of Songkla University, Hat Yai 90110, Songkhla, Thailand; siriluk.ka@psu.ac.th; 5Proteodiag Co., Ltd., Mueang Songkhla 90000, Songkhla, Thailand; 6Faculty of Medical Technology, Prince of Songkla University, Hat Yai 90110, Songkhla, Thailand; jidapa.sz@psu.ac.th; 7Division of Animal Production Innovation & Management, Faculty of Natural Resources, Prince of Songkla University, Hat Yai 90110, Songkhla, Thailand; chaiyawan.w@psu.ac.th

**Keywords:** lateral flow immunoassay (LFIA), *Burkholderia pseudomallei*, melioidosis, capsular polysaccharide

## Abstract

Early diagnosis is essential for the successful management of *Burkholderia pseudomallei* infection, but it cannot be achieved by the current gold standard culture technique. Therefore, this study aimed to develop a lateral flow immunoassay (LFIA) targeting *B. pseudomallei* capsular polysaccharide. The development was performed by varying nitrocellulose membrane reaction pads and chase buffers. The prototype LFIA is composed of Unisart CN95 and chase buffer containing tris-base, casein, and Surfactant 10G. The assay showed no cross-reactivity with *E. coli*, *S. aureus*, *P. aeruginosa,* and *P. acne.* The limit of detections (LODs) of the prototype LFIA was 10^7^ and 10^6^ CFU/mL *B. pseudomallei* in hemoculture medium and artificial urine, respectively. These LODs suggest that this prototype can detect melioidosis from positive hemoculture bottles but not straight from urine. Additionally, these LODs are still inferior compared to Active Melioidosis Detect (AMD^TM^). Overall, this prototype holds the potential to be used clinically with hemoculture bottles. However, further improvements should be considered, especially for use with urine samples.

## 1. Introduction

*Burkholderia pseudomallei*, an etiologic agent of melioidosis, is an aerobic, facultative intracellular, Gram-negative bacillus living in soil and water surfaces [1]. The bacterium can be transmitted to people through cutaneous inoculation, ingestion, or inhalation. *B. pseudomallei* is found in more than 48 tropical and subtropical countries around the world, with the highest prevalence in northern Australia and Southeast Asia, particularly Thailand [2]. The global incidence of melioidosis was estimated to be 165,000 cases, with approximately 89,000 deaths annually [3]. The global burden of melioidosis was estimated at 4.64 million disability-adjusted life-years [4].

Melioidosis is a neglected, life-threatening infection [5]. The mortality rate of melioidosis can be as high as 70% without medical intervention and 10% to 40% with antibiotic treatments [3,6]. Complications in the treatment of melioidosis are partly due to the intrinsic resistance of the bacterium to most antibiotics (e.g., penicillin, ampicillin, first- and second-generation cephalosporins, and aminoglycosides) [7]. In addition, a licensed vaccine for the prevention of melioidosis is currently unavailable. Due to its ability to cause a fatal infection through airborne transmission, intrinsic antibiotic resistance, and lack of a vaccine, *B. pseudomallei* is classified as a Tier 1 select agent by the Centers for Disease Control and Prevention (CDC) of the United States of America (USA), highlighting its significant threat to global health [8,9].

Melioidosis patients usually present with a wide spectrum of symptoms. Diagnosis of melioidosis based solely on clinical features is, therefore, difficult, making laboratory testing indispensable [10]. At present, bacterial culture from clinical specimens is still the gold standard for the diagnosis of the infection. This technique is only 60% sensitive and could take up to ten days to result, causing delays in the treatments [11,12]. Thus, rapid diagnostic tests are required to prompt the appropriate treatments and reduce the mortality risk.

In this study, we aimed to develop a rapid lateral flow immunoassay (LFIA) for the detection of *B. pseudomallei*. Principally, LFIA is a paper-based assay platform consisting of a sample pad, conjugate pad, reaction pad, and absorbance pad (also called a wicking pad) adhered to a backing card [13]. In an assay, the conjugate pad is sprayed with a detector antibody (mostly conjugated with gold nanoparticles), and the reaction pad is sprayed with a capture antibody to form a test line. To perform an assay, the sample is dropped on the sample pad and migrated to the conjugate pad, where the analyte binds to the detector antibody. The complex of analyte and detector antibody then migrates further to the reaction pad and interacts with the capture antibody, allowing the complex to concentrate and become visible on the test line [13]. Usually, it takes only 15–20 min to display the result. As compared to other rapid diagnostic techniques such as indirect hemagglutination (IHA), immunofluorescence assay (IFA), enzyme-linked immunosorbent assay (ELISA), and polymerase chain reaction (PCR), LFIA is found to satisfy most of the ASSURED (Affordable, Sensitive, Specific, User-friendly, Rapid and robust, Equipment-free, and Deliverable to end-users) criteria, set by the World Health Organization (WHO) for an ideal point-of-care (POC) diagnostic [14,15]. Based on these indicators, we selected the LFIA format for this research. We have noted that the antigen-capture LFIAs for *B. pseudomallei* detection have already been developed and commercialized by InBios International Inc., Washington, USA [16]. However, the test is not readily available in some endemic areas, such as Thailand. This limitation, therefore, urged us to develop the LFIA, starting from the isolation of our own monoclonal antibody to the development using materials and reagents commonly used by local manufacturers.

## 2. Materials and Methods

### 2.1. Bacteria

The use of the bacteria in this study was conducted in accordance with the Pathogens and Animal Toxins Act, B.E. 2558 (2015), and approved by the Institutional Biosafety Committee—Prince of Songkla University (IBC-PSU), approval number 007-2024. In the biosafety level 2 (BSL-2) laboratory, *Escherichia coli*, *Staphylococcus aureus*, *Pseudomonas aeruginosa,* and *Propionibacterium acne* were grown in Luria–Bertani (LB) broth at 37 °C for 24–48 h and inactivated at 70 °C for 3 h. The inactivation was verified by back-plating before use [17]. Heat-inactivated *B. pseudomallei* K96243 was received as a courtesy from the Center of Excellence Research for Melioidosis and Microorganisms, Walailak University, Nakhon Sri Thammarat, Thailand. Concentrations of all the bacteria were determined by measuring the optical density at 600 nm (OD_600_).

*B. pseudomallei* polysaccharide (Bp-PS) extract was prepared by mixing the inactivated bacterium with the Laemmli buffer and boiling for 10 min, followed by incubation with proteinase K (Life Technologies, Carlsbad, CA, USA) at 60 °C overnight. The sample was then dialyzed in purified water. The obtained Bp-PS were analyzed by SDS-PAGE with Coomassie blue and modified silver staining (Thermo Fisher Scientific, Grand Island, NY, USA).

### 2.2. Ethics Statement

The use of an animal laboratory in this study was carried out in accordance with the Animal for Scientific Purposes Act, A.D. 2015, and the recommendations in the Guide for the Care and Use of Laboratory Animals of the National Institutes of Health. The protocol was approved by the Institutional Animal Care and Use Committee, Prince of Songkla University (approval number 34/2021).

### 2.3. Immunization of Mice

A group of females, 8-week old, BALB/c mice (Nomura Siam International, Bangkok, Thailand) were immunized with 2 × 10^8^ colony forming units (CFU) of heat-inactivated *B. pseudomallei* K96243 in 200 μL of phosphate-buffered saline (PBS) via intraperitoneal injection, with subsequent boosts at weeks 2, 4, 6, and 8 [18]. Serum samples were collected via retro-orbital survival bleeds, and antibody titers were monitored by an indirect ELISA. Three days prior to splenectomy, a final boost of 2 × 10^8^ CFU of the inactivated bacterium was administered intraperitoneally. Mice were euthanized by CO_2_ asphyxiation, followed by cardiac punctures. Splenocytes were harvested from all mice and frozen until fusion.

### 2.4. Isolation of Monoclonal Antibodies

Hybridoma cell lines were generated by the standard hybridoma technique [19]. The splenocytes were fused with P3X63Ag8.653 (American Type Culture Collection, Manassas, VA, USA) using 50% polyethylene glycol, and the fused cells were grown in Dulbecco’s Modified Eagle’s Medium (DMEM) containing hypoxanthine-aminopterin-thymidine (HAT), fetal bovine serum (FBS), and P338D1 (American Type Culture Collection, Manassas, VA, USA) conditioned medium. Hybridoma clones producing *B. pseudomallei* antibodies were screened using an indirect ELISA. The selected clones were then subjected to multiple rounds of cloning by limiting dilution to ensure monoclonality. As a result, a hybridoma clone, Bp2.1, was isolated. Subsequently, the clone was adapted to a hybridoma serum-free medium (Thermo Fisher Scientific, Grand Island, NY, USA) for antibody production. Monoclonal antibody (mAb) Bp2.1 was purified from the hybridoma supernatant using recombinant protein A, which is known to have a binding affinity to the Fc region of various types of antibodies, including murine IgG [20,21]. Briefly, the supernatant containing Bp2.1 mAb was mixed with Binding buffer (20 mM sodium phosphate, pH 7.0) at a 1:1 ratio. The mixture was loaded into a HiTrap^TM^ rProtein A FF column (Cytiva, Marborough, MA, USA), followed by washing with the Binding buffer and eluting with 0.1 M sodium citrate, pH 3.0. Next, the purified mAb was passed through to a HiTrap^TM^ Desalting column (Cytiva, Marborough, MA, USA) in order to preserve the mAb in PBS. The concentration of the mAb was determined using OD_280_.

### 2.5. Indirect ELISA

Microtiter plates were coated overnight at 4 °C with 1 × 10^8^ CFU/mL of the inactivated *B. pseudomallei*. After washing with PBS containing 0.05% Tween-20 (PBS-T), the plates were blocked at 37 °C for 60 minutes with PBS containing 0.5% Tween-20 and 5% non-fat milk (blocking buffer). The plates were then incubated at room temperature for 90 minutes with samples diluted in a blocking buffer. After incubation, the plates were washed with blocking buffer and incubated with horseradish peroxidase (HRP)-labeled goat anti-mouse IgG antibody (Southern Biotech, Birmingham, AL, USA) at room temperature for 60 minutes, followed by washing with PBS-T. The analysis was also carried out with HRP-labeled goat anti-mouse IgG3, IgG1, IgG2a, and IgG2b antibodies (Southern Biotech, Birmingham, AL, USA) to determine the subclass of the isolated mAb. The plates were developed by adding tetramethylbenzidine (TMB) substrate (Thermo Fisher Scientific, Grand Island, NY, USA) into each well and stopped after 30 minutes with 1 M H_3_PO_4_. The OD_450_ was then read.

### 2.6. SDS-PAGE and Western Blot

Bp-PS extract was mixed with the Laemmli buffer, boiled for 10 minutes, and separated by 10% SDS-PAGE. After the separation, the polyacrylamide gels were subjected to Coomassie blue staining, modified silver staining, and western blotting. The silver staining started by fixing the gels in 0.65% periodic acid, 5% acetic acid, and 20% ethanol for 60 minutes to oxidize the carbohydrate [22]. Following the fixing step, the gels were stained using a silver staining kit (Thermo Fisher Scientific, Grand Island, NY, USA) (Appendix A). Western blotting was performed by the tank-blotting method. After blotting, the membranes were blocked with 5% skim milk in TBS-Tween (TBS-T: 50 mM Tris, 150 mM NaCl, 0.1% Tween 20, pH 7.6) at 4 °C overnight. Then, the membranes were probed with 5 μg/mL of Bp2.1 mAb for 90 minutes at room temperature. After washing with TBS-T, the membranes were incubated with the anti-mouse IgG HPR conjugate (Southern Biotech, Birmingham, AL, USA) at room temperature for 60 minutes and washed again with TBS-T. The membranes were developed using a chemiluminescent substrate. The results were recorded by the ImageQuan^TM^ LAS 500 imaging system (Cytiva, Marborough, MA, USA).

### 2.7. Artificial Urine

The artificial urine was prepared from the formula described by Shmaefsky et al. [23]. It consists of 303 mM urea, 128.3 mM sodium chloride, 60.4 mM potassium chloride, 40 mM sodium phosphate, 0.2% creatinine, 0.005% albumin, and a pH adjusted to 5.1.

### 2.8. LFIA

Initially, LFIAs were constructed with five different types of nitrocellulose membrane reaction pads provided in the Lateral Flow Materials Starter Kit (Set I) (Serve Science, Bangkok, Thailand). The set of five nitrocellulose membranes included Whatman FF170HP, FF120HP (Whatman, Buckinghamshire, UK), Unisart CN95, CN140, and CN140 unback (Sartorius, Göttingen, Germany). Each nitrocellulose membrane was glided with 0.8 mg/mL Bp2.1 mAb for a test line and 0.8 mg/mL goat anti-mouse IgG, unlabeled (Southern Biotech, Birmingham, AL, USA), for a control line. The gliding was carried out at a flow rate of 100 mm/second and with a volume of 1 μL/cm using a Kinbio reagent dispenser (Shanghai Kinbio Tech., Shanghai, China). After drying, the membranes were assembled with an Ahlstrom 8964 conjugate pad (Ahlstrom, Helsinki, Finland), a Millipore C048 sample pad (Merck KGaA, Darmstadt, Germany), and a Whatman 470 absorbance pad (Whatman, Buckinghamshire, UK), with each pad, overlapped by approximately 2 mm [16,24,25]. The assembled cards were then cut into 4 mm-width strips.

Bp2.1 mAb was conjugated to 40 nm colloidal gold nanoparticles (Kestrel BioSciences, Pathum Thani, Thailand) by passive absorption, blocked with PBS containing 10% bovine serum albumin, and concentrated to an OD_540_ of 10 [24,25]. The gold-conjugated Bp2.1 (4 μL) was added to the conjugate pad prior to running the assay to serve as a detection antibody. The assay was performed by loading 20 μL of sample onto the sample pad, followed by placing the strip into 200 μL of chase buffer. The suitable chase buffers were screened from the Running Buffer Screening Kit (Table 1) (Kestrel BioSciences, Pathum Thani, Thailand). The test and control lines were examined promptly after 20 minutes. The assays were interpreted as positive if both test and control lines were visible, and they were interpreted as negative if only the control line was visible. The assays were considered invalid if the control line was invisible and they were repeated. The ambiguous results were solved by three independent examiners reading the LFIAs in a blind manner [26].

The final prototype LFIA was constructed with a Unisart CN95 as a reaction pad. The control line of the final prototype was prepared with a goat anti-chicken IgY antibody (Southern Biotech, Birmingham, AL, USA). Gold-conjugated chicken IgY (Jackson ImmunoResearch, West Grove, PA, USA) was applied to the conjugate pad together with Bp2.1 gold conjugate prior to running the assays. The assays were carried out using KB-BS-0008 (Kestrel BioSciences, Pathum Thani, Thailand) as a chase buffer and interpreted as described previously. The limits of detection (LOD) of the assays were determined and assessed in comparisons with the Active Melioidosis Detect^TM^ Rapid Diagnostic Test or AMD^TM^ RDT (InBios, Seattle, WA, USA).

## 3. Results

### 3.1. Bp2.1 mAb

Bp2.1 is a mAb isolated from mice immunized with heat-killed *B. pseudomallei* using the standard hybridoma technique. The mAb Bp2.1 was purified from the hybridoma supernatant and characterized by western blotting and indirect ELISA (Figure 1). The western blot analysis was performed with Bp-PS blotted on the nitrocellulose membrane and probed with Bp2.1 mAb, followed by the goat anti-mouse IgG HRP conjugate. The result showed that Bp2.1 reacted to the antigen larger than 250 kDa (approximately 300 kDa), which corresponds to the size of the capsular polysaccharide (CPS) (Figure 1a) [26]. The indirect ELISA was carried out with the inactivated *B. pseudomallei* coated on the ELISA plates, followed by Bp2.1 as a primary antibody and the subclass-specific goat anti-mouse IgG3, IgG1, IgG2b, and IgG2a HRP conjugates as secondary antibodies. The results demonstrated that the mAb Bp2.1 is an IgG3 subclass (Figure 1b).

### 3.2. LFIA Prototype

An optimization of the LFIA was performed with five different types of nitrocellulose membranes and 15 different recipes of chasing buffers, generating a total of 75 conditions to be tested. First, the LFIAs were screened with 20 μL of PBS as a test sample (Figure 2). The LFIAs with Whatman FF170HP showed detectable test lines with all chasing buffers except buffer no. 1, in which an invalid test result was observed. These results were found to be similar to those from the Whatman FF120HP. Additionally, uneven control and test lines were observed for these two membranes. Similar results were observed for LFIAs with the Unisart CN140 unback. However, unlike the Whatman membranes, CN140 unback LFIAs (also CN95 and CN140) did not produce an invalid result with chasing buffer no. 1, and the control and test lines are smoother. 

For Unisart CN95 and CN140 LFIAs, there are five conditions (with chasing buffer no. 7, 8, 10–12) and two conditions (with chasing buffer no. 1 and 8), respectively, that did not yield false positive results, while the rest produced various degrees of background at test lines. Overall, a total of seven conditions that produced a clearly visible control line with an undetectable test line were selected. The selected conditions were subsequently tested with 20 μL of BacT/ALERT PF Plus hemoculture media (BioMérieux, Durham, NC, USA) (Figure 3). Out of the seven conditions selected, five LFIAs exhibited false positive results, leaving only two suitable LFIAs for the next step of development. Among these two, the LFIA with Unisart CN95 membrane and KB-BS-0008 (tris buffer containing casein and Surfactant 10G) chase buffer was chosen for the prototype because of the smoother flow pattern observed during the test. All other parameters and components used to construct the prototype LFIA are summarized in Table 2.

### 3.3. LOD

The LOD of the prototype LFIA was assessed using a fresh hemoculture medium spiked with different concentrations of the inactivated *B. pseudomallei*. The assays were inspected promptly at 20 min after placing the strips in the chasing buffer. The results were read as positive if both test and control lines were visible and negative if only the control line was visible. The LOD is defined as the lowest concentration of the inactivated *B. pseudomallei* that yields a positive result. Figure 4a demonstrates that our prototype LFIA had an LOD of 10^7^ CFU/mL, while the AMD^TM^ had an LOD of 10^6^ CFU/mL. We also determined the LOD of the prototype LFIA in the artificial urine matrix (Figure 4b). The results show that our prototype LFIA and the AMD^TM^ had LODs of 10^5^ CFU/mL and <10^5^ CFU/mL, respectively.

### 3.4. Cross-Reactivity

The potential cross-reactivity to other bacteria was tested. The bacterial panel tested included *S. aureus* and *P. acne* as representatives of Gram-positive bacteria and *E. coli* and *P. aeruginosa* as representatives of Gram-negative bacteria. The tests were performed with the killed bacteria spiked in the hemoculture medium matrix (Figure 5a) and the urine matrix (Figure 5b) at a final concentration of 10^8^ CFU/mL. In the two matrices tested, the prototype LFIA showed no cross-reactivity with these bacteria.

## 4. Discussion

It is well established that an early diagnosis is essential for the successful management of melioidosis by triggering timely treatment, which in turn reduces the mortality risk [10]. In the current study, we describe the development of the prototype antigen capture LFIA for the detection of *B. pseudomallei*. An antigen detection-based assay was chosen over serology tests because serological assays cannot detect an early infection in which the specific antibody is not yet present. We have acknowledged that the antigen captures LFIA for melioidosis detection has already been developed and commercialized by InBios International Inc., under the trade name AMD^TM^ [16]. However, the product is not readily available in some countries, and it has not yet been approved for clinical use.

The development of an antigen capture LFIA in the current work started with the immunization of mice with killed *B. pseudomallei* to isolate mAbs specific to the bacterium. The obtained mAb Bp2.1 was found to be an IgG3 subclass, which is a predominant subclass of mouse IgG in response to polysaccharide antigens (Figure 1b) [27,28]. The target antigen of Bp2.1 is *B. pseudomallei* capsular polysaccharide (CPS) (Figure 1a). CPS is a major antigen located on the bacterial outer membrane, which is easily accessible by antibodies. Its structure is common among *B. pseudomallei* but differs from those of other species except *B. mallei* [29]. These characteristics make CPS a suitable target for melioidosis detection. The development of the LFIA, therefore, proceeded with the CPS-specific mAb Bp2.1. We have also noted that CPS is also targeted by AMD^TM^ RDT [16]. Additionally, CPS contains repeating epitopes, allowing Bp2.1 to be used as both a capture and detector antibody.

Once the target molecule and antibody were designated, the LFIAs were assembled. The assay construction was performed using materials and buffers that are commonly used or made by local manufacturers in the country to avoid disruption in the event that scaling-up is needed. During the optimization, we decided to primarily focus on the nitrocellulose membrane and chase buffer, the two major components that usually affect the test accuracy [30,31]. The other components of the LFIAs were selected based on the previous study and were kept unchanged throughout the optimization [25]. With five different nitrocellulose membranes and 15 chase buffers, we created a total of 75 LFIA conditions to be examined. The initial screening was carried out using PBS as a sample (Figure 2). The LFIAs run with tris-based chase buffers (KB-BS-0007 to 0012) did not exhibit false positive results regardless of the types of nitrocellulose membrane used, suggesting that the tris-based buffer system is more suitable for our LFIAs as compared to PBS-based (KB-BS-0001 to 0006) and citrate-phosphate-based (KB-BS-0015) buffer systems. Casein in tris-based buffers seemed to be essential for suppressing non-specific binding and preventing a false positive result (KB-BS-0007 and 0008 versus KB-BS-0013 and 0014). We observed that KB-BS-0001 (PBS containing Surfynol^®^ 465) flowed very slowly and could not reach the absorbance pad within 20 min, causing invisible or faint control lines, especially with Whatman FF120HP and FF170HP. During the optimization, we also noticed that the nitrocellulose membranes with a slow flow rate tended to exhibit non-specific binding at 20 min, and a longer running time (>20 min) was required for the test lines to disappear. This finding, however, is different from the results reported by previous studies in which the assay sensitivity increased as a flow rate decreased due to an increase in contact time at a test line [32,33,34]. Additionally, we found that test and control lines were sharper on the Unisart membranes, even though all the membranes were handled similarly. Based on those results, seven LFIAs that (i) showed a clearly visible control line, (ii) did not yield false positive results at 20 minutes, (iii) flowed evenly, and (iv) had sharp test and control lines were selected for further optimization (Figure 2, asterisked).

Currently, melioidosis is diagnosed by a culture technique, which could be performed either by culturing the specimens (e.g., blood, urine, pus) directly on selective media such as Ashdown’s agar or by growing bacteria using a hemoculture bottle followed by identification with selective media or other techniques [10,35,36]. Thus, we first tested the selected LFIAs with a normal serum. All the selected conditions exhibited false positive results, indicating that our LFIAs were not suitable to be used directly with serum. Then, we tested the LFIAs with a hemoculture medium (Figure 3). Of the seven conditions tested, there were only two that did not yield a false positive result. Among these, the LFIA with Unisart CN95 and KB-BS-0008 buffer (tris buffer containing casein and Surfactant 10G) was chosen due to a smoother flow pattern. Additionally, chicken IgY gold conjugate and anti-chicken IgY were employed to enhance the control line intensity. Consequently, the prototype LFIA developed in this study consists of a Unisart CN95 reaction pad sprayed with Bp2.1 mAb at the test line and anti-chicken IgY at the control line, Bp2.1 gold conjugate, chicken IgY gold conjugate, KB-BS-0008 chase buffer, and other components as listed in Table 2.

After the prototype LFIA was developed, its LODs were investigated to understand the performance of the assay. We found that our prototype LFIA had the LOD of 1 × 10^7^ CFU/mL of the inactivated *B. pseudomallei* in the hemoculture matrix (Figure 4a). Previous studies reported that a positive hemoculture bottle contained 2 × 10^7^ to 7 × 10^9^ CFU/mL of bacteria, which was higher than the LOD of our LFIA [37,38]. This indicated that, with the procedure illustrated in Appendix A, the prototype LFIA has the potential to detect *B. pseudomallei* directly from a positive hemoculture bottle, just like the AMD^TM^ RDT reported previously [39]. 

During the infection, *B. pseudomallei* presents not only in the patient’s bloodstream but also in urine [12,40]. A study performed by Wuthiekanun et al. reported that the median concentration of *B. pseudomallei* in urine samples was 1.5 × 10^4^ CFU/mL, with the highest and lowest concentrations of ≥1 × 10^6^ CFU/mL and ≤1 × 10^3^ CFU/mL, respectively [40]. In this study, the prototype LFIA’s LOD was 1 × 10^6^ CFU/mL in artificial urine (Figure 4b). This LOD value suggested that the ability of the LFIA to detect the bacterium straight from urine samples was only marginal, and its sensitivity must be improved.

In comparison with the LFIA developed previously, it is important to stress that our prototype LFIA had LODs higher than those of AMD^TM^ RDT. AMD^TM^ sensitivity was approximately 10 times greater than our LFIA. However, in the present work, we only optimized two parameters contributing to the assay performance. It should be noted that this prototype must be optimized further, for example, by investigating other factors systematically, such as membrane treatment, antibody concentration, particle size of gold, etc. [30]. A step of sample preparation, e.g., centrifugation, may be introduced to help improve the sensitivity of the assay [12].

Lastly, the prototype LFIA was investigated for potential cross-reactivity with other bacteria. This characteristic is primarily governed by the specificity of antibodies. Our LFIA was developed using *B. pseudomallei* CPS-specific mAb Bp2.1; thus, its potential cross-reactivity would be deduced from other mAbs specific to the same target. The previous works have illustrated that *B. pseudomallei* CPS-specific mAbs were not reactive with other bacteria, including *B. pseudomallei* near-neighbor species, except *B. mallei* and particular strains of *B. cepacia* [16,41,42]. Therefore, it is reasonable to deduce that the prototype LFIA would cross-react with *B. mallei* and *B. cepacia* but not with other pathogens. In the present study, we also tested the LFIA prototype with *S. aureus, P. acne*, *E. coli*, and *P. aeruginosa,* which represent both Gram-positive and Gram-negative bacteria. The results showed no cross-reactivity (Figure 5). However, the future study should include a larger panel of bacteria for cross-reactivity investigation.

## 5. Conclusions

The LFIA for *B. pseudomallei* detection has been developed successfully using the newly isolated CPS-specific mAb Bp2.1. The prototype assay had the LOD of 1 × 10^7^ CFU/mL in hemoculture medium, which is sensitive enough to detect *B. pseudomallei* in positive hemoculture bottles. In the artificial urine matrix, the LFIA exhibited the LOD of 1 × 10^6^ CFU/mL *B. pseudomallei*. It is not sensitive enough to reliably detect melioidosis in urine samples. The performance of the prototype LFIA is slightly inferior compared to the AMD^TM^. Altogether, it was suggested that the LFIA prototype has the potential to advance to clinical applications, especially with hemoculture bottles. However, to be used with other types of samples, such as urine, more optimizations must be performed.

## Figures and Tables

**Figure 1 diagnostics-14-01033-f001:**
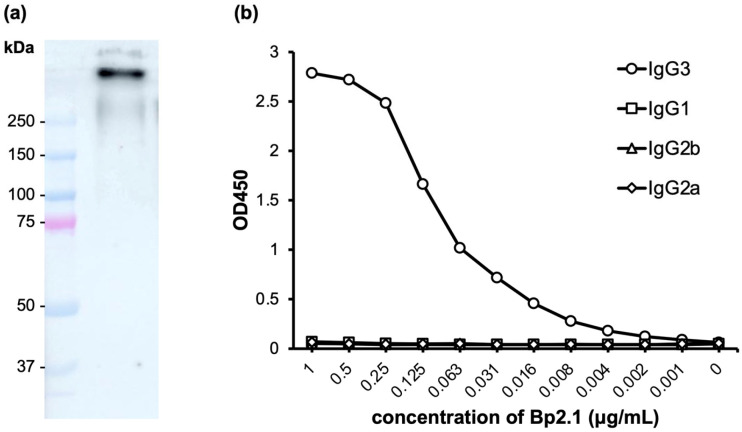
**Characterization of the mAb Bp2.1 using Western blotting (a) and indirect ELISA (b).** Western blot analysis (**a**) was carried out with proteinase K-treated *B. pseudomallei* lysate (Bp-PS) separated on 10% SDS-PAGE gel, blotted on the nitrocellulose membrane, and probed with Bp2.1 mAb. The result shows the interaction between Bp2.1 and a high molecular antigen with the size corresponding to CPS. An indirect ELISA (**b**) with the heat-killed *B. pseudomallei* immobilized on the plates, and Bp2.1 as a primary antibody shows the interaction with anti-mouse IgG3 HRP. The result indicates that Bp2.1 is an IgG3 subclass.

**Figure 2 diagnostics-14-01033-f002:**
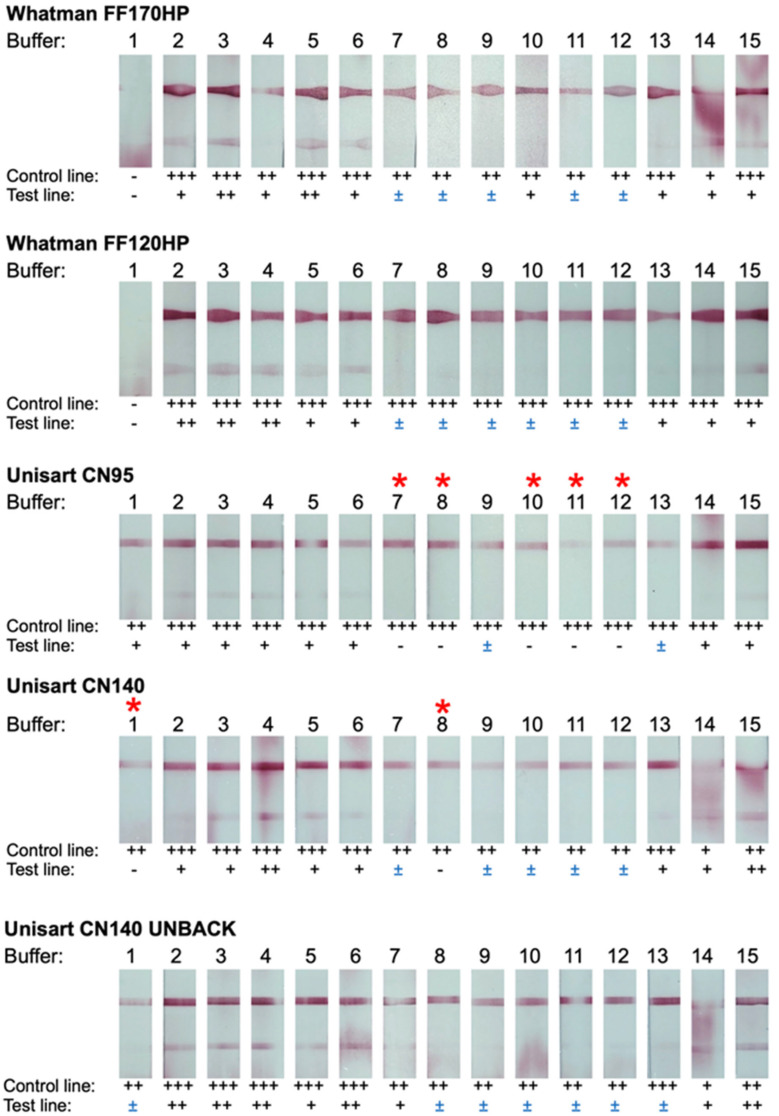
**Membranes and chase buffers screening for the LFIA development.** The optimal conditions for the LFIAs were searched for by varying nitrocellulose membrane types and chase buffer formulas (no. 1-15, Table 1). PBS was used as a sample for the screening. The intensity of control and test lines was visually inspected and recorded as “+++”, “++”, “+”, and “−” for high, moderate, low, and no band intensity, respectively. “±” marks the conditions that require > 20 min for the test lines to disappear. The optimal conditions, which are the conditions that produced the visible control line with the invisible test line, were asterisked (*).

**Figure 3 diagnostics-14-01033-f003:**
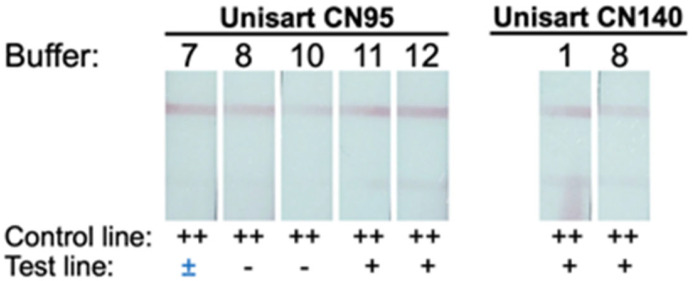
**LFIAs testing with a hemoculture medium.** The selected conditions from the previous screening were tested with fresh hemoculture medium, and the test results were inspected visually at 20 minutes. The intensity of control and test lines was recorded as “++”, “+”, and “−” for moderate, low, and no band intensity, respectively. “±” marks the conditions that require > 20 min for the test lines to disappear. The testing identifies two conditions that did not produce false positive results (Unisart CN95 with buffer no. 8 and Unisart CN95 with buffer no. 10, Table 1).

**Figure 4 diagnostics-14-01033-f004:**
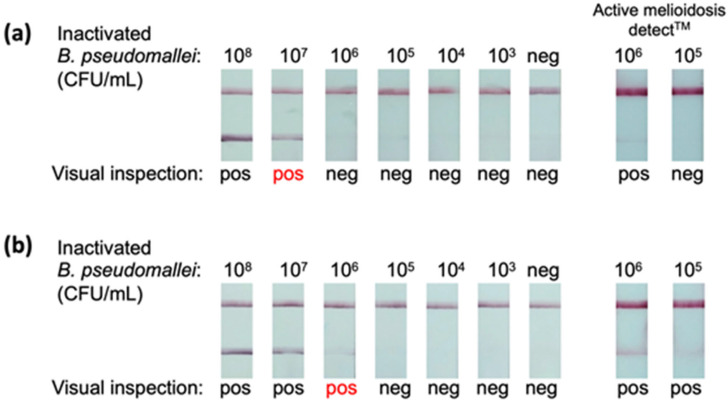
**Determinations of the LODs of the prototype LFIA using heat-killed *B. pseudomallei.*** The LODs of the prototype LFIA were determined using fresh hemoculture medium (**a**) and artificial urine (**b**) spiked with various concentrations of the heat-killed bacterium. The Active Melioidosis Detect^TM^ RDTs were also included for comparison. The lowest bacterial concentrations giving a positive result (indicated by a red label) marked the LOD values. The results illustrate that the LODs for the heat-killed *B. pseudomallei* in fresh hemoculture medium and artificial urine were 10^7^ CFU/mL and 10^6^ CFU/mL, respectively. (Pos = positive, Neg = negative).

**Figure 5 diagnostics-14-01033-f005:**
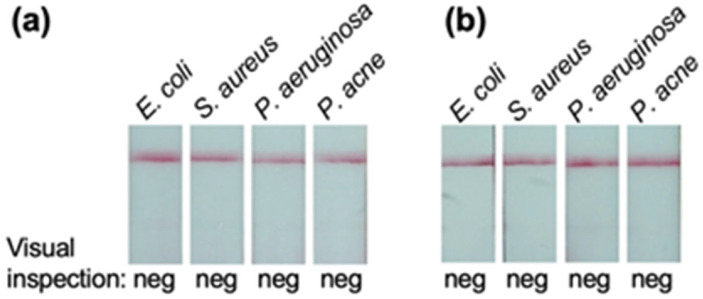
**Cross-reactivity assessment.** The potential cross-reactivity of the LFIA was assessed using killed *S. aureus* and *P. acne* (Gram-positive representatives) and *E. coli* and *P. aeruginosa* (Gram-negative representatives) spiked in the hemoculture medium (**a**), and the artificial urine (**b**). The prototype LFIA did not exhibit any cross-reactivity with these bacteria under the investigated matrices. (Pos = positive, Neg = negative).

**Table 1 diagnostics-14-01033-t001:** List of chase buffers included in the Running Buffer Screening Kit.

No.	Code	Components
1	KB-BS-0001	PBS + Surfynol 465
2	KB-BS-0002	PBS + Surfactant 10G
3	KB-BS-0003	PBS + Tween-80
4	KB-BS-0004	PBS + Tween-20
5	KB-BS-0005	PBS + Triton X-405
6	KB-BS-0006	PBS + Triton X-100
7	KB-BS-0007	Tris base + Casein + Surfynol 465
8	KB-BS-0008	Tris base + Casein + Surfactant 10G
9	KB-BS-0009	Tris base + Casein + Tween-80
10	KB-BS-0010	Tris base + Casein + Tween-20
11	KB-BS-0011	Tris base + Casein + Triton X-405
12	KB-BS-0012	Tris base + Casein + Triton X-100
13	KB-BS-0013	Tris base + Triton X-405
14	KB-BS-0014	Tris base + Tween-20
15	KB-BS-0015	Citrate-phosphate buffer pH 5.5

**Table 2 diagnostics-14-01033-t002:** Summary of parameters and components for the LFIA prototype assembly.

Parameters/Components	
Reaction pad:	Unisart CN95
Conjugate pad:	Ahlstrom 8964
Sample pad:	Millipore C048
Absorbance pad:	Whatman 470
Control line:	Goat anti-mouse IgY, 0.32 μg/strip
Test line:	Bp2.1 mouse IgG3, 0.32 μg/strip
Flow rate:	100 mm/sec
Conjugates:	Bp2.1 mAb gold conjugate plus chicken IgY gold conjugate
Chase buffer:	KB-BS-008
Strip width:	4 mm

## Data Availability

Data supporting the reported results may be provided upon reasonable request to the corresponding author.

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
