# Peer review of "Development of an Antigen Capture Lateral Flow Immunoassay for the Detection of Burkholderia pseudomallei"

_diagnostics, 2024, doi:10.3390/diagnostics14101033_

Round 1

Reviewer 1 Report

Comments and Suggestions for Authors

The manuscirpt # diagnostics-2955626 deals with the development of the prototyğe antigen capture Lateral  Flow immunoassay (LFI) for the detection of Burkholderia pseudomallei. The topic is big essential  and deserves publication. I would recommend its publication in the journal after following issues are addressed.

 Lines 61-64.. "In this study, we aimed to develop a rapid lateral flow immunoassay (LFI) for the detection of B. pseudomallei. LFI was selected among other rapid diagnostic techniques (e.g., indirect hemagglutination (IHA), immunofluorescence assay (IFA), enzyme-linked immunosorbent assay (ELISA), and polymerase chain reaction (PCR)) because it is simpler, cheaper, and faster, with only 20 – 30 minutes to display the result."...

This part needs revision regarding more information on LFI with current refs. should be given.

Lines 114-119.. "As a result, a hybridoma clone, Bp2.1, was isolated. Subsequently, the clone was adapted to hybridoma serum-free medium (Thermo Fisher Scientific) for antibody production. Monoclonal antibody (mAb) Bp2.1 was purified from the hybridoma supernatant using protein A affinity column chromatography (Cytiva) and stored in PBS. The concentration of the mAb was determined using OD280."   

The authors nowhere describe how they realize protein A affinity column chromatography. This part should elaborated.

Lines 211-215.. "An optimization of the LFI was performed with five different types of nitrocellulose membranes and 15 different recipes of chasing buffers, generating a total of 75 conditions to be tested. First, the LFIs were screened with 20 L of PBS as a test sample, and the conditions that produced a clearly visible control line with an undetectable test line were selected (Figure 2)."..

More discussion with current refs on Fig 2, including each panel should be given. 

Lines 242-246.. "The results were read as positive if both test and control lines were visible, and negative if only the control line was visible. The LOD is defined as the lowest concentration of the inactivated B. pseudomallei that yields a positive result. Figure 4a demonstrates that our prototype LFI had the LOD of 107 CFU/mL, while the AMDTM had the LOD of 106 CFU/mL. We also determined the LOD of the prototype LFI in the artificial urine matrix (Figure 4b)."..

This part needs the comparision with current promising approaches, including LOD values.  

Comments on the Quality of English Language

minor revision is needed.

Reviewer 2 Report

Comments and Suggestions for Authors

This paper “Development of an antigen capture lateral flow immunoassay 2 for the detection of Burkholderia pseudomallei” developed a lateral flow immunoassay targeting B.pseudomallei capsular polysaccharide, which was performed by varying nitrocellulose membrane reaction pads and chase buffer. LFA is not new technology, thus, I think that this research is very meaningful from the clinical diagnosis. I suggested to publishing this paper after minor revisions.

1.      I think that author should add some result of clinical samples.

Reviewer 3 Report

Comments and Suggestions for Authors

The study is devoted to the development of LFIA for the determination of Burkholderia pseudomallei.

The authors note that the developed assay is not unique, because today there is a commercially available analogue (10 times more sensitive). However, the authors also point out that the commercial test is not available in some countries. Please describe why.

Questions also arise regarding the development strategy. For example, the authors claim that they obtained monoclonal antibodies to the polysaccharide, using killed bacterial cells for immunization. Is it not better to use pure polysaccharide for these purposes? How do the authors ensure that the resulting antibodies do not bind to other cell surface structures?

In this case, a single antibody clone is used in a sandwich assay. That is, this test is only capable of detecting analytes with repeating epitopes. Perhaps the use of two clones would increase the sensitivity of the assay, which this test lacks.

Reviewer 4 Report

Comments and Suggestions for Authors

«Development of an antigen capture lateral flow immunoassay 2 for the detection of Burkholderia pseudomallei» from Thailand is excellent work showing the development of a rapid test for the presence of bacteria. Given its wide habitat, the development of such tests is extremely important.

The development approach is impressive. Various types of nitrocellulose paper have been tested. This is a good replacement for lengthy selection of block buffers. The group obtained the antibodies independently and showed good affinity. In general, the work was done at an extremely high level and is distinguished by its scale. The work may be presented in its original form.

The method has not reached real implementation, but it has already shown very good experience with input and found and has shown insufficient sensitivity, but has the potential for improvement with the help of buffers.

Four minor comments below

- Lateral flow immunoassay usually has the abbreviations LFIA or LFT from Lateral flow test

- I would like to see a diagram of the test system and a diagram of sample preparation.

- Figure one shows 2 times

- Please rejuvenate the links. Most of the material is a little outdated

Round 2

Reviewer 1 Report

Comments and Suggestions for Authors

The authors addressed all my concerns and the revised manuscirpt is now suitable for publication.

Comments on the Quality of English Language

minor revision is needed.

Reviewer 3 Report

Comments and Suggestions for Authors

The authors provided detailed comments on all questions. The manuscript can now be published in the 'Diagnostics' journal.